# Development of New Probe-Based Real-Time RT-qPCR Assays for the Detection of All Known Strains of Bovine Ephemeral Fever Viruses

**DOI:** 10.3390/v17030407

**Published:** 2025-03-13

**Authors:** Natalia Golender, Eyal Klement, Bernd Hoffmann

**Affiliations:** 1Koret School of Veterinary Medicine, The Robert H. Smith Faculty of Agriculture, Food & Environment, The Hebrew University of Jerusalem, P.O. Box 12, Rehovot 76100, Israel; golendernat@gmail.com; 2Department of Virology, Kimron Veterinary Institute, Bet Dagan 5025001, Israel; eyal.klement@mail.huji.ac.il; 3Institute of Diagnostic Virology, Friedrich-Loeffler-Institut, Südufer 10, 17493 Greifswald-Insel Riems, Germany

**Keywords:** cattle, outbreak, epizootic, *Ephemerovirus*, *Rhabdoviridae*, TaqMan, sensitivity, specificity, laboratory diagnosis, quantitative real-time PCR (qPCR)

## Abstract

Bovine ephemeral fever is an arthropod-borne viral disease that affects cattle and buffalo in many regions of the world; it causes heavy economic losses in the cattle industry. To date, all BEFV-specific diagnostic molecular assays have been based on the variable glycoprotein (G-protein)-coding genome region, potentially allowing the pathogen to escape detection. We developed two new assays, based on the less variable nucleoprotein genome region, and compared them with two G-protein-based assays. For this comparison, we used 245 samples comprising positive and negative field samples from Israeli outbreaks caused by different strains, belonging to lineage I and IIIa, as well as Australian and Japanese strains (lineages IV and IIIb). The new assays showed high agreement with the previous assay (Kappa = 0.92), detecting 144 out of 147 positive samples (sensitivity of 97.96%), and detected 6 more samples as positive out of 98 samples found negative by the G-protein-based assay. All nine non-agreeing results were validated as positive using a conventional RT-PCR assay. The new assays have higher analytical sensitivity than the previous assays, can be combined with internal controls, and enable the detection of all known BEFVs. The results indicate that these two nucleoprotein-based real-time RT-qPCRs can serve as fast, sensitive, and specific assays for the sustainable detection of BEFV strains.

## 1. Introduction

Bovine ephemeral fever (BEF) is an arthropod-borne viral disease of cattle and buffalo [1]. BEFV is suspected to be transmitted by some genera of mosquitos (Diptera: Culicidae), such as *Anopheles*, *Culex*, *Uranotaenia*, and *Aedes*, as well as several different species of biting midges *Culicoides* (Diptera: Ceratopogonidae) [2].

The morbidity rate in susceptible cattle populations is usually high (potentially reaching 80–100%), whilst the mortality in most cases remains at about 1–2% [3]. The duration of clinical signs is usually short, characterized by bi-phasic acute hyperthermia, lameness, and ocular and nasal discharge before recovery [2]. In more seriously affected cattle, recumbency, synovitis, lethargy, muscle stiffness, lameness, reluctance to move, inappetence, anorexia, and death may be reported. Moreover, neurotropism caused by BEFV has been confirmed through histopathological examinations of the brain, spinal cord, and peripheral nerves; this may causally contribute to ataxia, paresis, or paralysis [4,5]. The major economic impact caused by BEF in cattle is attributed to reduced milk production, death, and the culling of seriously affected animals in the dairy industry; meanwhile, in the feedlot cattle industry, economic damage is linked to a loss of production, temporary infertility in males, and the disabling of draught animals [1]. The disease has been reported in Africa, Asia, Australia, and the Far and Middle East [3,6,7,8,9,10,11,12]. In endemic areas, the disease has been reported every two to four years [6,10]. In Israel, disease outbreaks were recorded in 2009–2010, 2014–2015, 2018, 2021, and 2023. In 2023, an outbreak was caused by novel “Mayotte-like” strains, which were new to the region [10,13].

BEFV (species *Ephemerovirus febris*) belongs to the *Rhabdoviridae* family and the *Ephemerovirus* genus. The non-segmented, negative-sense single-stranded RNA ~14.9 kb long genome of the BEFV virion encodes five structural proteins (nucleoprotein (N), phosphoprotein (P), matrix protein (M), glycoprotein (G) and RNA-dependent RNA polymerase (L)) and five non-structural proteins (NSs) (Gns, α-1, α-2(3), β, γ) [14,15]. The BEFV G protein is the target of virus-neutralizing antibodies [3], and, like all viral outer proteins, it is the most variable viral protein. Based on a phylogenetic analysis of the available G-coding regions of the BEFV genome, all BEF viruses belong to a single serotype, which is further divided into four lineages (I–IV) [10]. In contrast, N and L are the most conservative proteins among all ephemeroviruses. This allows for the development of group-specific N- and L-based molecular methods for the detection of viruses belonging to the genus *Ephemerovirus* [16,17].

The modern methods of BEFV detection and diagnosis are commonly based on real-time quantitative reverse transcription PCR (RT-qPCR). Several different types of RT-qPCR systems have been developed. They include a loop-mediated isothermal amplification (RT-LAMP) assay, lateral-flow dipstick isothermal recombinase PCR assays, SYBR green I–based qPCR assays, conventional RT-PCR assays, and probe-based RT-qPCR assays [17,18,19,20,21,22,23]. All these methods are based on the detection of the genome regions of the G protein of BEFV strains.

Due to the high mutation and diversity rates of G-coding genome sequences, PCR methods of BEFV detection may fail to detect emerging strains. In the current study, we developed and validated two new fluorescence-labeled probes for N-protein-based RT-qPCR assays, which were designed according to the latest available BEFV sequence information. These new RT-qPCR assays were tested in parallel with two RT-qPCR assays, which were previously used for the laboratory diagnosis of BEFVs in the Kimron Veterinary Institute, Israel [13,21]. Both of the new BEFV-RT-qPCR assays were equipped with internal controls (IC) and were validated using a comprehensive validation panel comprising 245 samples, of which 98 specimens pretested negative and 147 pretested positive for BEFV according to the previously used method [13,17].

## 2. Materials and Methods

### 2.1. Samples

For this study, we tested a total of 245 field samples of bovine origin, which were collected from routine field sample submissions for the molecular diagnosis of arboviral infections at the Kimron Veterinary Institute. These samples included 147 BEFV-positive samples (60%) collected from BEFV outbreaks between 2009 and 2023; 94 samples from outbreaks caused by BEFV strains of lineage IIIa; 52 samples from lineage I; and 1 RNA sample extracted from a vial of an inactivated vaccine based on Japanese BEFV strain K-KB (lineage IIIb). The 98 negative field samples (40%) were mostly collected during the last BEFV outbreak between 2023 and the beginning of 2024. Information about the number and the type of samples is presented in Table 1. These samples were pretested using two methods: a conventional group-specific method for ephemeroviruses RT-PCR [17], with or without re-testing using the BEFV-specific RT-qPCR assay described by Erster et al. [13]. The final “positive–negative” evaluation of the samples was conducted according to the assay developed by Erster et al. [13], as shown in Table 1.

### 2.2. Viral RNA Extraction

Ribonucleic acid (RNA) from the cell culture supernatant/vaccine and field samples (whole blood in EDTA, lung, brain, and spleen) was extracted using a MagMAX™ CORE Nucleic Acid Purification Kit (Thermo Fisher Scientific, Austin, TX, USA) with KingFisher Duo Prime (Thermo Fisher Scientific, Vantaa, Finland), or using an IndiMag Pathogen Kit (Indical Bioscience, Leipzig, Germany) with IndiMag 48 (INDICAL BIOSCIENCE, Leipzig, Germany) according to the instructions of the manufacturer.

To extract RNA from the tissue/organs of the dead animals, tissues were homogenized, diluted with PBS at a proportion of 1:10, and centrifugated to remove large tissue debris; the supernatant was used for the consequent RNA extraction.

### 2.3. Real-Time RT-qPCR Assay Preparation

To improve the reliability of BEFV diagnosis, two BEFV-RT-qPCRs were designed. Two completely new probe-based RT-qPCR assays (BEFV-N-Mix-2 and BEFV-N-Mix-8) were selected as the most sensitive out of an initial set of 12 primer–probe combinations that were located on heterogeneous genetic locations of the N gene within the BEFV genome. The primers and TaqMan probes were designed based on an alignment of 21 BEFV nucleoprotein genome sequences that were available on the NCBI database in August 2023 (MN148803; MN148801; MN148802; MN148800; KM276084; MW463337; MW512963; MN905763; KY315724; MN078236; MH756623; MH105245; KY012742; KC688889; U04166; KC688890; OP887034, MZ687779; JX234572; JQ941694; JQ941691). Both BEFV-RT-qPCR assays were performed under the same conditions with a total reaction volume of 12.5 μL. To enhance the security of the new RT-qPCR systems, they were validated as duplex one-step RT-qPCRs using the double-check strategy in combination with internal control (IC) systems based on the detection of an endogenous IC (ß-Actin-Mix 5-HEX/JOE) [24,25] (Table 2).

#### 2.3.1. Probe–Primer Mix Preparation

Stocks of primers and probes were resuspended by adding TE buffer (pH 8.0) to a final concentration of 100 μM/μL according to the recommendations of the manufacturer. BEFV-specific and internal control mixes of the primers and probes were prepared according to Table 3.

#### 2.3.2. Master Mix Preparation

At the initial validation stage, five different RT-qPCR kits were used for validation: qScript™ XLT 1-Step RT-qPCR ToughMix^®^ (Quantabio, Beverly, MA, USA), AgPath-ID^TM^ One-Step RT-PCR Kit (Life technologies, Austin, TX, USA), Clara™ Probe 1-Step Mix No-ROX (PCR biosystems, London, UK), GoTaq 1-Step RT-qPCR System (Promega, Madison, WI, USA), and the One Step PrimeScript™ RT-PCR Kit (Takara Bio Inc., Kusatsu, Japan). The major validation tests were conducted using three kits, qScript™ XLT 1-Step RT-qPCR ToughMix^®^, AgPath-IDTM One-Step RT-PCR Kit, and Clara™ Probe 1-Step Mix No-ROX, due to their superior preliminary results. To perform the tests published by Stram et al. [21], a qScript™ XLT 1-Step RT-qPCR ToughMix^®^ kit was used. For the assay developed by Erster et al. [13], the SensiFAST SYBR^®^ No-ROX Kit (Bioline, London, UK) was used according to the published protocols. Information about the master mix preparation is shown in Table 4.

The temperature/time profiles for all five used kits are shown in Table 5. The temperature/time profiles were generated according to the recommendations of the commercial kits and were maintained using a QuantStudio™ 5 Real-Time PCR Instrument (Thermo Fisher Scientific/Applied Biosystems, Waltham, MA, USA).

### 2.4. Repeatability (Reproducible Quality)

Repeatability was determined via threefold RNA extractions of four positive field samples: three whole-blood samples collected in 2021 (A and B) and 2023 (C) and one washed buffy coat sample, which was collected in 2018 (D). All tests were performed in duplicate. Comparative tests within the samples were performed using the systems developed by Stram et al. [21] and Erster et al. [13] and with recently developed systems (BEFV-N-Mix-2 and BEFV-N-Mix-8) using three commercial kits (qScript XLT One-Step RT-qPCR ToughMix, AgPath-ID™ One-Step RT-PCR Kit, and Clara™ Probe 1-Step Mix No-ROX).

### 2.5. Analytical Sensitivity

Analytical sensitivity was determined using the limit-of-detection method (LOD) based on spiking on the confirmed negative bovine whole-blood sample of two BEFV isolates: Australian BB7721 (AUS, 10^7.47^ TCID_50_/mL, accession number (acc. no.) U04166) and the Israeli Mayotte-like isolate ISR-1512/23 (10^7.3^ TCID_50_/mL acc. no. PQ409321). Local Israeli BEFV strains (lineage IIIa) were not used for the LOD determination due to their complete inability to be detected by the Stram et al. assay [21]. Since RNA extraction was conducted using 0.1 mL of every BEFV isolate, the initial concentration for the AUS strain was taken as 10^6.47^ of infection particles and as 10^6.30^ for the ISR-1512/23. Infectious particles were translated from the TCID_50_/mL titer into plaque assay infectious units per milliliter (IU/mL) [26]. IU were calculated by the conversion of 100 μL of extracted RNA to 2.5 μL used per reaction. The samples were tested from undiluted (supernatant of the infected Vero cells) and log_10_ dilutional series (10^−1^ to 10^−6^), corresponding to 5164.6 and 3491.7, 516.5 and 349.2, 51.7 and 34.9, 5.2 and 3.5, 0.52 and 0.35, and 0.052 and 0.035 IU, respectively, for the AUS and ISR-1512/23 strains. All tests were performed similarly to the method described in Section 2.4. If PCR product amplification was observed in both replicates, the detected dilution was used to determine the LOD.

### 2.6. Analytical Specificity

The inclusivity (detection of samples containing related target organisms) and exclusivity (detection of only the unique target organism but no cross-reaction to related target organisms) [27] of the BEFV-specific RT-qPCR assays were tested on several available BEFV strains, belonging to three out of four BEFV lineages. These were lineages I, IIIa, IIIb, and IV (n = 4), a recently identified ephemerovirus in Israel (a Hefer Valley virus (HVV) [17] (HVV, n = 2)), and other common and relevant cattle arboviruses found in Israel and around the world, including bluetongue virus (BTV) serotypes 1, 3, 4, 5, 6, 8, 9, 11, 12, 15, and 16 (n = 11); epizootic hemorrhagic disease virus (EHDV) serotypes 1, 6, and 7 (n = 4); and other relevant bovine viruses such as the bovine respiratory syncytial virus (BRSV, n = 1), parainfluenza virus 3 (PIV-3; n = 1), malignant catarrhal fever (MCFV; n = 1), bovine herpes viruses 1 and 4 (BHV-1; n = 1; BEV-4; n = 1), the Akabane virus (AKAV; n = 1), the Shuni virus (SHUV; n = 1), and the Schmallenberg virus (SBV; n = 1). Several bovine bacterial and blood parasitic pathogens causing symptoms resembling those of BEF were also used in the analytical testing, including Babesia bigemina (n = 1), Theileria annulata (n = 1), Anaplasma marginale (n = 1), Salmonella D (n = 1), Leptospira interrogans serogroup Pomona (n = 1), and Pasteurella multocida serotype B (n = 1). All tests were performed similarly to the method described in Section 2.4.

### 2.7. Diagnostic Sensitivity and Specificity

For further validation of the methods used to evaluate diagnostic sensitivity and specificity, a comprehensive sample panel containing BEFV-positive specimens (n = 147) and negative specimens (n = 98) was used (see Section 2.1). All samples of the validation panel were tested using the two novel BEFV-RT-qPCR assays (Mix-2 and Mix-8). Diagnostic sensitivity and specificity were calculated in accordance with the two-by-two table [28,29].

The newly developed BEFV-RT-qPCR assays were compared with BEFV-specific RT-qPCR systems previously used in the Division of Virology of the Kimron Veterinary Institute; these were developed by Erster et al. [13] and Stram et al. [21]. The system developed by Stram was also tested as a duplex RT-qPCR with the same IC (ß-Actin-Mix 5-HEX); the primer–probe premix was prepared in the same way as recently developed BEFV-specific assays, and the premix was designated as BEFV-G-Mix-1 (Table 2).

## 3. Results and Discussion

### 3.1. Reproducible RNA Extraction from Cattle Field Samples

Triplicates of the RNA extractions for each of the four samples (samples A–D, Table 6) yielded very similar results, which are labeled (1) to (3). Each RNA extraction was tested in duplicate, and the average of the duplicates is presented in Table 6. The raw data for this reproducible quality test of the RNA extracted from positive field samples are shown in the Appendix A.

According to the results, three out of the four samples, which belong to lineage IIIa of BEFV [10], were not detected by the assay developed by Stram et al. [21]. In contrast, sample C, which belongs to lineage I, was amplified by all four types of RT-qPCR assay. The assay developed by Erster et al. [13], which is based on the SYBR Green I type of RT-qPCR, detected all samples and achieved amplification during an earlier cycle compared to all of the other BEFV-specific assays used for the comparison (Table 6). Nevertheless, the Sybr Green RT-qPCR has the disadvantage of not using an additional probe to confirm the specificity of the assay. Furthermore, it should be noted that the SensiFAST SYBR^®^ No-ROX Kit is known for its very early Ct values. However, these very early Ct values do not necessarily indicate increased analytical sensitivity. This can only be determined by comparing the corresponding dilution series.

### 3.2. Analytical Sensitivity (The Limit of Detection (LOD))

The LODs, shown as standard curves, are illustrated in Figure 1 and are based on the inoculation of a negative bovine whole-blood sample with two different BEFV strains of lineage I (ISR-1512/23) and lineage IV (AUS—Australian strain BB7721). Figure 1a presents consolidated LOD data for all four systems tested with the AUS BEFV strain, while Figure 1b presents consolidated LOD data for all four systems tested with the ISR-1512/23 BEFV strain. The LOD of the assay developed by Stram et al. [21] shows about 5 infectious particles for the AUS and 3.5 – for the ISR-1512/23 and BEFV strains. For the assay developed by Erster et al. [13], the limit of detection was about 5 IU for the AUS strain and 35 IU for the ISR-1512/23 strain. Regarding the recently developed BEFV-specific Mix-2 and Mix-8 assays, the Mix-2 assay showed detection of 5 and 3.5 IU for the AUS and ISR-1512/23 BEFV strains, respectively. Mix-8 detected 5 IU of the AUS strain and about 0.35 IU of the ISR-1512/23 strain. Since the new assays showed similar results to all three selected RT-qPCR kits (see Appendix A), we present the data from only one kit (AgPath-ID™ One-Step RT-PCR Kit) in Figure 1, as its amplified PCR product was detected a few cycles earlier than with the other two RT-qPCR kits. Raw data about the dilution series based on water are also presented in Appendix A. According to the raw data from the dilution series based on water and whole-blood samples, water-based dilution showed no background in the Erster et al. assay [13]. As a result, it exhibited no false-negative results and higher sensitivity compared to the test using spiked whole-blood samples (even when considering the estimated results). Since routine RNA extraction uses field samples without pretreatment, this led to a high number of false-negative results from the samples with Ct values lower than 30. Compared to the repeatability results, all Ct values of all samples were lower than Ct 30 for the Erster et al. system, and, as a consequence, no false-negative results were observed.

### 3.3. Analytical Specificity

The inclusivity of the test was 100%, demonstrating the detection of all tested BEFV lineages: I, IIIa, IIIb, and IV. No amplification was observed for other tested microorganisms that can cause infections in cattle resembling BEFV infection, illustrating the excellent exclusivity of the newly developed BEFV-specific RT-qPCR tests.

### 3.4. Diagnostic Sensitivity (DSe) and Diagnostic Specificity (SPe)

The results of the 43 field samples tested in duplicate are shown in Appendix A. The tests were performed using the assays developed by Stram et al. [21] and Erster et al. [13], while the recently developed Mix-2 and Mix-8 methods were tested using five different commercial kits (the names of the kits and the time/temperature modes are listed in Table 5). The results of 202 field samples, which were tested in a single replicate, are presented in Appendix A. These tests were run using Stram et al. [21] and Erster et al. [13], and the recently developed Mix-2 and Mix-8 methods were tested using the three different commercial kits that provided the best results: Quanta (qScript XLT One-Step RT-qPCR ToughMix (Quantabio)); AgPath (AgPath-ID™ One-Step RT-PCR Kit (Life technologies)); and Clara (Clara™ Probe 1-Step Mix No-ROX (PCR biosystems)). The same positive and the same negative samples, which were defined as positive or negative using the Erster et al. [13] assay, showed 100% agreement between both of the recently developed BEFV-specific RT-qPCR assays with all three commercial kits. The new assays detected 144 out of the 147 samples found positive by Erster et al. The DSe of both assays was determined to be 97.96% (CI95% = 94.27–99.3%), while six out of 98 negative samples were also found to be positive by the two new assays. In total, the Mix-2 and Mix-8 assays detected 150 positives and 95 negatives out of 245 tested samples. The agreement between the assay developed by Erster et al. and the two newly developed assays was very high (Kappa = 0.92, *p* < 0.0001). Nine samples, which showed no agreement between the assays developed by Erster et al. [13] and the Mix-2 and Mix-8 assays, were tested using a pan-ephemero conventional RT-PCR [17], and all of them were shown to be positive.

## 4. Conclusions

In summary, the new BEFV assays (BEFV-N-Mix-2 and -8) detected all of the tested BEFV strains, including strains from lineage IIIa, which were not detected using the system of Stram et al. [21]. Furthermore, the new tests demonstrated better analytical sensitivity than the test developed by Stram et al., with lower Ct values observed for lineage I BEFV strains (‘Mayotte-like’ strains).

The assay developed by Erster et al. [13] provides lower Ct values when testing washed, pretreated buffy coat samples, as compared to the newly developed assays; it also detects all strongly positive samples, regardless of lineage. However, non-pretreated field samples and samples with low viral burdens and high background rates could be evaluated as “false negative” by the Erster et al. assay, while the same samples were determined to be positive by both the Mix-2 and Mix-8 assays. Additionally, the absence of an internal control makes it impossible to assess the quality of the samples using the Erster et al. assay. This could also lead to false results.

Sequences from the glycoprotein (G) gene are frequently used in the molecular epidemiology of BEFV. However, the sequencing protocol for this G-region carries a risk of contamination in diagnostic BEFV screening assays. In contrast, The N-gene-based BEFV assays presented here operate independently and are less prone to contamination. This makes the new tests more practical for routine BEFV diagnosis.

Both newly developed BEFV-N-based assays can be used in combination with a housekeeping gene (beta-actin)-based internal control as a duplex PCR. This enables the detection of inhibitory effects and helps to avoid false-negative results. Given the proven robustness of the assays, combining them with other established internal control systems is also feasible.

Due to the generally quite high genetic variability of both the known and unknown BEFV strains circulating around the world, two BEFV assays with approximately equal sensitivity and specificity were developed based on the relatively conserved N gene to ensure reliable molecular diagnoses. The two BEFV-N-Mix-2 and -8 assays are genetically independent, allowing them to be used in a double-check strategy, which is particularly useful for new outbreaks. Using two independent PCR systems enhances diagnostic reliability and reduces the likelihood of false-negative results. A comparison of the two assays showed that the BEFV-specific N-based Mix-8 assay generally had a slightly higher BEFV detection rate than the Mix-2 assay. The systems can be used separately, but, for the confirmation of uncertain results, we recommend using both systems.

## Figures and Tables

**Figure 1 viruses-17-00407-f001:**
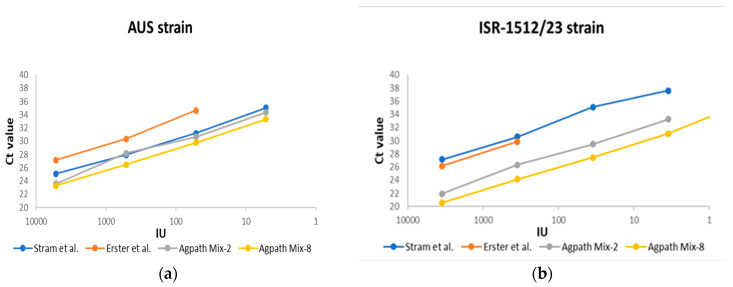
Standard curves of the dilution series of the two BEFV strains tested using different BEFV-specific RT-qPCRs. The BEFV isolates from Australia (BB7721-AUS) and Israel (ISR-1512/23) were tested using both the Stram assay [21] and the Erster et al. SYBR Green assay [13], as well as the new BEFV-N-Mixes 2 and 8. The results for the AgPath-ID™ One-Step RT-PCR Kit are shown here. The raw data, including the results with Quanta (qScript XLT One-Step RT-qPCR ToughMix kit) and Clara (Clara™ Probe 1-Step Mix No-ROX kit), are summarized in Appendix A. IU—infectious units. (**a**) United data for the analytical sensitivity tests of all four systems tested using the AUS BEFV strain; (**b**) united data for the analytical sensitivity tests of all four systems tested using the ISR-1512/23 BEFV strain.

**Table 1 viruses-17-00407-t001:** Information about the number and type of samples used for the validation tests.

Sample	W.blood	B.coat	Serum/Plasma	Spleen	S+L	Brain	Inact.vac.	Total
№ pos	55	81	4	3	3	0	1	147
№ neg	70	13	4	6	2	3	0	98
Total	125	94	8	9	5	3	1	245

W.blood—whole blood in EDTA; B.coat—buffy coat; S+L—mixed equally according to the weights of the lung and spleen samples; Inact.vac.—inactivated vaccine.

**Table 2 viruses-17-00407-t002:** Information about recently developed and previously published real-time RT-qPCR assays used in the present study for the detection of all known BEFV genomes and internal controls.

PCR Assay	Designation of Oligo	Sequence of Oligo 5′–3′	Amplicon Length	Location	Source
BEFV-	BEFV-N-46-F	GTSTTTYAACAGGTCTCTTTCCT	79	N gene	Current
N-Mix-2-	BEFV-N-124-R	TCAGTTGGCTTAACAGCCTTG			study
	BEFV-N-102-FAM	FAM-TCTCTTTCTTRTTCAATGTGCART ACAT-BHQ1			
BEFV-	BEFV-N-311-F	GGAACTTTTATGARGTAATCATAGA	85	N gene	Current
N-Mix-8	BEFV-N-395-R	CATACATCATCATTYTCATCYACATT			study
	BEFV-N-363-FAM	FAM-CTTCTHCATCTATTGTCTGRTCATC-BHQ1			
beta-	ACT-1030-F	AGCGCAAGTACTCCGTGTG	106	beta-	[25]
actin-	ACT-R-1135-R	CGGACTCATCGTACTCCTGCTT		actin-	[24]
Mix5	ACT-PROBE-HEX	HEX-TCGCTGTCCACCTTCCAGCAGATGT-BHQ2		gene	[24]
SYBR	BEFV-G-1140F	GAATCATTATGGGATMGGATC		G gene	[17]
	BEFV-G-1273R	CCTCCTGCTGGTGCTGTTTC			
BEFV-	BEFV-G-70F	GAGATCAAATGTCCACAACGTTTAA	73	G gene	[21]
G-Mix-1	BEFV-G-142R	AATGTTCATCCTTTGCAAGATTATGA			
	BEFV-G-98FAM	FAM-AATTATCACTTCAAGCCC-MGB			

**Table 3 viruses-17-00407-t003:** Composition of primer–probe premixes for BEFV-N-Mix-2 and-8 and BEFV-G-Mix-1.

	BEFV-Mix-1/-2/-8	Beta-Actin-Mix-5
Primer F	10 μL	2.5 μL
Primer R	10 μL	2.5 μL
Probe	2.5 μL	2.5 μL
Water	77.5 μL	92.5 μL
Total	100 μL	100 μL

Mix-1—premix of BEFV-G-Mix-1; Mix-2/-8—premixes of BEFV-N-Mix-2 and-8 (Table 2).

**Table 4 viruses-17-00407-t004:** Master mix preparation using the different commercial kits.

	Commercial Kit				
Reagents	Quanta	AgPath	Clara	Takara	Promega
Buffer	6.25 μL	6.25 μL	3.12 μL	6.25 μL	6.25 μL
RT	-	0.5 μL	-	0.25 μL	0.25 μL
EX Tag HS	-	-	-	0.25 μL	-
BEFV-N-Mix-2/-8 *	1 μL	1 μL	1 μL	1 μL	1 μL
Beta-actin-Mix-5 *	1 μL	1 μL	1 μL	1 μL	1 μL
Water	1.75 μL	1.25 μL	4.88 μL	1.25 μL	1.5 μL
Total	10 μL	10 μL	10 μL	10 μL	10 μL

*—premixes prepared according to Table 3; RT—reverse transcriptase; Quanta—qScript XLT One-Step RT-qPCR ToughMix (Quantabio); AgPath—AgPath-ID™ One-Step RT-PCR Kit (Life technologies); Clara—Clara™ Probe 1-Step Mix No-ROX (PCR biosystems); Promega—GoTaq 1-Step RT-qPCR System (Promega); Takara—One Step PrimeScript™ RT-PCR Kit (Takara Bio Inc.).

**Table 5 viruses-17-00407-t005:** Temperature/time profiles for every commercial kit used in the study.

		Commercial Kit				
Cycle	Step	Quanta	AgPath	Clara	TaKaRa	Promega
1x	RT	10 min/50 °C	10 min/45 °C	10 min/52 °C	5 min/45 °C	15 min/45 °C
1x	denaturation	2 min/95 °C	10 min/95 °C	3 min/95 °C	30 s/95 °C	2 min/95 °C
	denaturation	10 s/95 °C	15 s/95 °C	15 s/95 °C	5 s/95 °C	15 s/95 °C
45x	annealing	15 s/58 °C	30 s/58 °C	15 s/58 °C	15 s/58 °C	15 s/58 °C
	elongation	15 s/68 °C	45 s/68 °C	30 s/68 °C	34 s/68 °C	45 s/68 °C

RT—reverse transcription step; Quanta—qScript XLT One-Step RT-qPCR ToughMix (Quantabio); AgPath—AgPath-ID™ One-Step RT-PCR Kit; Clara—Clara™ Probe 1-Step Mix No-ROX (PCR biosystems); Promega—GoTaq 1-Step RT-qPCR System (Promega); Takara—One Step PrimeScript™ RT-PCR Kit (Takara Bio Inc.).

**Table 6 viruses-17-00407-t006:** Reproducible quality of the RNA extracted from positive field samples.

									Assay Type/RT-qPCR Kit								
		Stram/Quanta		Erster/ Sensi		Quanta		Quanta		AgPath		AgPath		Clara		Clara	
Sample (Replicate)	Source	Ct	β-ACT	Ct	Tm	MIX2	β-ACT	MIX8	β-ACT	MIX 2	β-ACT	MIX 8	β-ACT	MIX 2	β-ACT	MIX 8	β-ACT
A(1)	w.b.	NA	28.97	27.66	78.36	34.12	30.42	32.63	31.25	31.28	27.38	30.87	28.06	32.57	29.07	30.81	29.33
A(2)	w.b.	NA	28.69	26.67	78.05	33.58	30.55	33.06	32.26	31.17	27.24	30.35	28.04	31.97	29.01	30.73	29.84
A(3)	w.b.	NA	29.51	26.15	78.28	33.1	30.84	31.76	31.81	30.63	27.98	29.85	27.95	31.82	29.93	30.28	30.5
B(1)	w.b.	NA	30.49	29.49	78.28	35.77	32.00	34.82	32.86	33.51	28.94	33.16	31.14	35.19	30.66	32.64	31.88
B(2)	w.b.	NA	30.31	29.99	78.36	36.76	31.6	35.66	32.49	32.95	29.07	32.97	29.79	36.02	31.99	32.62	32.22
B(3)	w.b.	NA	30.57	28.8	78.2	35.08	32.18	33.75	33.41	33.97	31.53	32.32	30.26	34.54	31.74	32.05	31.73
C(1)	w.b.	26.63	26.49	15.28	80.02	22.08	27.78	19.78	28.92	19.71	25.79	18.47	26.08	19.93	27.58	18.75	28.47
C(2)	w.b.	26.30	26.48	15.57	79.94	22.06	28.58	20.17	28.62	20.00	26.28	18.80	26.07	20.05	26.97	19.02	27.83
C(3)	w.b.	25.43	26.44	14.15	79.80	20.52	28.31	18.08	27.35	18.59	25.55	16.98	25.88	19.02	26.70	17.67	27.44
D(1)	b.coat	NA	26.23	22.18	78.05	26.57	27.82	24.71	28.44	23.96	24.78	23.20	25.17	25.03	26.63	23.07	27.31
D(2)	b.coat	NA	25.66	22.01	78.05	26.29	27.33	24.91	28.43	23.96	24.60	26.34	27.05	24.28	25.44	23.22	26.31
D(3)	b.coat	NA	26.06	22.27	78.05	27.03	28.28	25.64	28.95	24.33	24.74	23.47	24.44	24.94	26.26	23.44	26.82

Stram—assay designed by Stram et al. [21]; Erster—assay designed by Erster et al. [13]; Sensi—SensiFAST SYBR® No-ROX Kit (Bioline); all the last columns show comparative tests of Mix-2 and Mix-8 according to recently validated assays; Ct—cycle threshold; w.b.—whole blood; b.coat—buffy coat; NA—not amplified; Tm—melting temperature; β-ACT—β-actin; Quanta—qScript XLT One-Step RT-qPCR ToughMix (Quantabio); AgPath—AgPath-ID™ One-Step RT-PCR Kit (Life technologies); Clara—Clara™ Probe 1-Step Mix No-ROX (PCR biosystems).

## Data Availability

All data are presented in the text and in the Appendix A.

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
