# Peer review of "Development of New Probe-Based Real-Time RT-qPCR Assays for the Detection of All Known Strains of Bovine Ephemeral Fever Viruses"

_viruses, 2025, doi:10.3390/v17030407_

Round 1

Reviewer 1 Report

Comments and Suggestions for Authors

The study aims to standardize and validate two RT-qPCR protocols for the diagnosis of Bovine Ephemeral Fever. The manuscript is well-written, demonstrating care and attention to detail, such as in the table captions. However, a thorough revision of the English language is necessary, as there are many missing prepositions and mispositioned adjectives.

Here are some suggested revisions before publication:

Lines 49-51: This paragraph is misplaced and should be moved to line 35 as a continuation of the first sentence of the introduction.

Lines 85 and 89: Where did the outbreaks occur? This information would be important to include.

Lines 116-117: Provide the accession numbers of the 21 genomic sequences used.

Lines 166-167 and 177-178: The reference numbering is incorrect.

Line 208: The text should mention Section 2.1.

Lines 242-262: Consider placing this section after the "Analytical Sensitivity" topic to maintain the logical flow of the methodology description.

Line 264: The entire "Analytical Sensitivity" section should come before "Diagnostic Sensitivity and Specificity."

Lines 273-274: There is repetitive text that needs to be corrected.

Supplementary tables: They should be renumbered according to their new order in the text.

Where are the results for "Analytical Specificity"? The methodology is presented, but the corresponding results are missing.

In summary, this is a well-written manuscript, but it requires an English revision and some adjustments before publication.

Comments on the Quality of English Language

A thorough revision of the English language is necessary, as there are many missing prepositions and mispositioned adjectives.

Reviewer 2 Report

Comments and Suggestions for Authors

The authors have developed two real-time RT-qPCR assays to detect different strains of bovine ephemeral fever virus. The methods are described quite in details; six tables and a figure are added.  

Apart from the thorough revision of English (preferentially by a native speaker) I advise the authors to do also some corrections in the quantification of the detected pathogens.

P5, L172-183: I assume that the TCID50/ml of both viruses (AUS and ISR) were determined in a tissue culture infectious dose 50 assay before. However, TCID50/ml is not the same as the number of infectious virions in the culture and it is not the same as number of genome copies. Stram and Erster in their publications quantified their assays with the use of the RNA transcript which can help to quantify genome copies/well or µl or ml or whichever volume is chosen. The Figure 1 (a), b), c), d)) describes the y axis as log10 (TCID50/well) which is quite confusing as those authors did not use the TCID50 value.

The whole chapter 2.5. Analytical Sensitivity should be carefully revised and the authors should clarify the exact method they used for the quantification of their two assays. If the TCID50/ml is used for concentration determination, it should be converted to infectious units (IU/ml) properly as those two characteristics are not the same.

P7, L267-268: “The LOD of the Stram et al. showed about one infectious particle for both ISR and AUS…“ That is not correct. The quantification of Stram at al. did not use quantification in infectious particles and did not show LOD as one particle. Stram et al. used the RNA transcript and converted the mass of RNA (ng) to the number of RNA molecules. I am citing “the sensitivity of the reaction is equivalent to about 10–100 viral genome molecules”. Please, correct the text accordingly.

Comments on the Quality of English Language

P1, L18: Please, correct “codding” (“coding” is the right expression).

P2, L58: “Basing of phylogenetic analysis of available G coding regions of the BEFV genome” is not comprehensible, please, reword.

P2, L62: "...methods for detection viruses..." Please, correct ("...methods for detection of viruses...").

P2, L64: "Several different types RT-qPCR..." A preposition is missing ("...types of RT-qPCR...").

P2, L73: “…were designed base on…” Please, reword. (e.g. “were designed according to …”).

P2, L87: “…which basing on Japanese BEFV strain…” Please, rephrase for more clarity.

P3, L107: “…according instructions…” Please, correct (“…according to the instructions…”)

P3, L108: “died animals” - should be “dead animals”.

P3, L118: The whole sentence in not comprehensible, please, correct (assays were carried out in parallel with what?)

P5, L165: “dublicates” – please, correct a typo (should be “duplicates”).

P5, L177: “…for the reason of complete inability being detected by Stram et al…” Please, reword to increase the clarity of the text.

P5, L180-1: “In case of the PCR product amplification was registered in both plicates, this indicator of the detected dilution was used for determination of the LOD.” Please, rephrase this sentence. The word “plicate” cannot be used in this connotation. Instead, the word “replicate” should be used. The same applies to the Line 247 – “single plicate” is not English. Use “single replicate” instead.

Round 2

Reviewer 2 Report

Comments and Suggestions for Authors

The authors corrected the manuscript according to comments, however, there is still an inaccuracy in the methodology (the calculation of Anylytical sensitivity - LOD). 

P5, L181-190: According to the protocols cited in the chapter References (citation No. 26), the calculation of IU/ml from TCID50/ml  is done as follows:

"Assume the conditions used for plaque assay and TCID assay don't alter the expression of infectious virus, TCID50/ml and pfu/ml are related by

pfu/ml = 0.7 * TCID50.

The concentration of the two undiluted strains would then be 1.397x105 (BB7721 AUS strain) and 1.01x105(ISR-1512/23) IU/ml.  After the series of dilution (10-1 to 10-6), the numbers correspond to 13970 (10100), 1397 (1010), 139.7 (101), 14 (10), 1.4 (1.0) and 0.14 (0.1) IU/ml. This differs almost by 2log10 when compared to the data shown in the article. Then the whole quantification of the assay is misleading. After working with RNA viruses for many years, I seriously doubt that it is possible to detect 0.5 IU in RT-qPCR. 

RNA extraction from 0.1 ml of undiluted virus does not change the virus concentration as long as the extraction method used elutes RNA in the same volume (0.1 ml). Then the concentration stays the same (e.g. 1.397x10IU/ml for AUS strain). If the authors choose to quantify the virus for the PCR reaction in one well, then the concentration must be converted to the numbers of virus RNA genomes in an amount in one reaction (e.g. 1 ul) which then would be 1.397x10(number of IU of AUS strain in 1 ul).

The whole chapter 2.5. Analytical Sensitivity should be once more carefully revised.
